# Forecasting shipping index using CEEMD-PSO-BiLSTM model

**Chengang Li[1,2], Xuan Wang[1,3], Yongxiang Hu[ORCID][4]\*, Ying Yan[1], Han Jin[1], Guofei Shang[4]**

**1** School of Big Data Application and Economics, Guizhou University of Finance and Economics, Guiyang, Guizhou, China, **2** Guizhou Key Laboratory of Big Data Statistics Analysis, Guizhou University of Finance and Economics, Guiyang, Guizhou, China, **3** New Structure Financial Research Center, Guizhou University of Finance and Economics, Guiyang, Guizhou, China, **4** School of Land Science and Space Planning, Hebei GEO University, Shijiazhuang, Hebei, China

\* doctorhu110@hgu.edu.cn

**Data Availability Statement:** All relevant data are within the paper and its Supporting Information files.

**Funding:** Our funding came from Hebei GEO University and Guizhou University of Finance and

## Abstract

Shipping indices are extremely volatile, non-stationary, unstructured and non-linear, and more difficult to forecast than other common financial time series. Based on the idea of "decomposition-reconstruction-integration", this article puts forward a combined forecasting model CEEMD-PSO-BiLSTM for shipping index, which overcomes the linearity limitation of traditional models. CEEMD is used to decompose the original sequence into several IMF components and RES sequences, and the IMF components are recombined by reconstruction. Each sub-sequence is predicted and analyzed by PSO-BiLSTM neural network, and finally the predicted value of the original sequence is obtained by summing up the predicted values of each sub-sequence. Using six major shipping indices in China's shipping market such as FDI and BDI as test data, a systematic comparison test is conducted between the CEEMD-PSO-BiLSTM model and other mainstream time-series models in terms of forecasting effects. The results show that the model outperforms other models in all indicators, indicating its universality in different shipping markets. The research results of this article can deepen and improve the understanding of shipping indices, and also have important implications for risk management and decision management in the shipping market.

## 1. Introduction

Since modern times, international shipping has become an important part of world trade and economic exchanges between countries. Compared with other means of transportation, ship transportation has the advantages of large carrying capacity and low operating costs. Today, more than 90% of the world's trade is completed by sea. The shipping index, as a price index constructed by the actual shipping market rates, not only comprehensively reflects the level of shipping rates in the maritime transportation market, but also objectively reflects the degree of fluctuation in the shipping market, and reflects the economic trend of a country to a certain extent. With the development of shipping market and the continuous growth of world trade volume, a country's economy and shipping trade demand are getting closer and closer. More and more scholars link shipping index with global economic development trend [1–3], among which BDI and other shipping indices have been used as economic indicators of world trade

Economics and two projects. Two projects: Key Project of Philosophy and Social Science Planning in Guizhou Province (No.: 20GZD61); 2020 Graduate program of Guizhou University of Finance and Economics (No.: 2020ZXSY09). The sponsor is Chengang Li and Xuan Wang respectively. The role of Chengang Li in the research is as follows: Conceptualization, Funding acquisition and Supervision. The role of Xuan Wang in the research is as follows: Funding acquisition, writing-original draft, writing-Review & editing.

**Competing interests:** The authors have declared that no competing interests exist.

by various countries [4]. Many experts and scholars at home and abroad have been working on how to grasp the trend of future changes in the shipping market through the prediction of shipping indices. Based on a reasonable forecasting model, investors can get excess returns from it, and managers and decision makers can guide the company's strategic decision-making according to the future trend, thus effectively avoiding market risks.

Previous studies have pointed out that shipping indices are non-linear, highly noisy and periodic [5–7], and in recent years, with the development of big data, a series of breakthroughs have been made in the field of machine learning and deep learning, which are widely used in the field of time series forecasting. The concept of deep learning, first proposed by Hinton et al. [8], is a new type of multi-hidden layer neural network based on artificial neural networks (ANN), which has shown powerful capabilities in the extraction of essential features of complex input samples. However, as the shipping market itself is closely linked to the international economic forms and macroeconomic cycles, resulting in shipping indices with characteristics such as cyclicality, multi-dimensionality and high complexity, if the deep learning model is directly applied to the prediction of time series, the model will highly fit the complex and chaotic noise in the original series during the training process, resulting in a weakened generalization ability of the model, and thus unable to learn and fit the original series of key features. To solve this problem, many scholars have tried to explain the characteristics of financial time series from signal decomposition, among which the empirical modal decomposition proposed by Huang et al. [9] and its modified model effectively decompose the high-frequency components and low-frequency components of financial time series, which enables the prediction accuracy of neural network models to be continuously improved.

This article adopts the idea of "decomposition, reconstruction and integration" to construct a comprehensive model—CEEMD-PSO-BiLSTM model, aiming to analyze and predict the internal characteristics and trend of shipping index, grasp the dynamic trend of shipping market, and prevent the major risks that may be brought by shipping market. At the same time, the prediction effect of deep learning in shipping index is explored to fill the gap of neural network in the field of shipping index prediction. The contribution of this article are as follows: (1) We introduce the neural network algorithm system into the shipping market prediction, and through the current better CEEMD method, the original sequence is decomposed, which creates favorable conditions for accurately fitting the nonlinear and high noise characteristics of shipping index, and significantly reduces the difficulty of neural network prediction. (2) We choose BiLSTM model with strong generalization ability in deep learning models as the framework, and combine BiLSTM model with CEEMD model in the field of signal decomposition to build a high-precision combined prediction model based on shipping index, providing a practical and reliable modeling scheme. (3) PSO was introduced to optimize the BiLSTM model, which significantly improved the prediction accuracy of shipping index.

The specific structure of this article is as follows: "Related works" section illustrates the rationality of this article through previous research work; "Models" section introduces the sub-models used in this article and the construction process of the CEEMD-PSO-BiLSTM combined model; "Empirical Analysis" section uses the CEEMD-PSO-BiLSTM model to predict and analyze the shipping index; "Results and discussion" section will compare other single models and combined models to illustrate the effectiveness of the model proposed in this article; "Summary" section is related research conclusions.

## 2. Related works

As the shipping market is always full of uncertain events, the quantitative analysis method of ocean freight index has always been the focus of international scholars. Since modern times,

more and more literatures have put forward the methods of freight index prediction. With the development of computer technology, the analysis of shipping index in academic circles is becoming more and more active for traditional model of freight index, such as ARMA model, ARIMA model, GARCH model, VAR model and so on. Cullinane et al. [10] conducted a univariate time series prediction analysis on Baltic Freight Index (BFI), and they successfully demonstrated that ARIMA model has a good effect in short-term prediction of BFI. Veenstra and Frans [11] established a vector autoregressive model for marine dry bulk freight samples and showed that there is a stable long-term relationship between different series of freight rates. Li [12] established ARMA model after eliminating trend and seasonal factors, which further improved the short-term prediction ability of BFI. Yang et al. [13] selected four vessel types of the Baltic Dry Index to build a GARCH (1,1) model, which exactly reflected the sensitivity and persistence pattern of its fluctuation. Chen et al. [14] used ARIMA and VAR models to predict spot rates of dry bulk routes of three major routes respectively, and they found that the VAR model had significantly better prediction effect than ARIMA model. Traditional models, although well constructed theoretically, must satisfy their prescribed statistical assumptions before a statistical model can be built, and they do not apply to high-latitude, noise-laden time series. When dealing with complex time series such as shipping data, traditional econometric models are unable to learn and fit their non-smooth, non-linear characteristics, resulting in unsatisfactory forecasting results.

With the advent of the Big Data era, the surge of data volume and the breakthrough development of computer computing power, the gradual rise of the artificial intelligence industry has led to new solutions to the financial time series forecasting problem. The emergence of machine learning has gradually replaced the use of traditional econometric models in financial time series. Yang et al. [15] selected CCBFI, CCFI and BFI as early warning criteria and used SVM models to successfully measure the prediction interval of the degree of freight alarm. Cho and Lin [16] used a fuzzy neural network model to analyze and forecast the BDI. The results showed that the fuzzy neural network model had high forecasting accuracy. Pérez-Cruz et al. [17] showed that when forecasting the return volatility of the stock market, the SVM model was used to estimate the parameters of the GARCH model, and this estimation method had higher forecasting ability than the ordinary ML method. Liu et al. [3] proposed an AR-SVR-GARCH model and an AR-SVR-GJR model. The empirical results show that both models have better volatility forecasting ability for the dry bulk shipping market, crude oil shipping market and shipping stock market. Neural networks (ANN), a multilayer neural network-based machine learning algorithm, has received considerable attention in the industry due to its suitability for solving complexity problems. Li and Parsons [18] showed that neural networks can significantly outperform time series models, especially in long-term forecasting. Kamal et al. [19] used deep neural networks to transform the BDI forecasting problem into a high-dimensional multiple regression problem. Zeng et al. [20] showed that the proposed ANN method outperformed VAR in the study of BDI forecasting. Sahin et al. [21] showed that the ANN method is an important modelling and forecasting method in the field of BDI forecasting and demonstrated its applicability. Zhang et al. [22] showed that ANN-based algorithms yielded less error and higher directional matching rates than econometric algorithms when forecasting weekly and monthly data models.

Schmidhuber and Hochreiter put forward the LSTM networks [23], which, as an improvement to the RNN model, has significantly better prediction results on time series than the traditional RNN model, due to its addition of memory units and forgetting units in the network structure, making it overcome the gradient dissipation or gradient explosion problem. In the study of time series prediction, many papers have shown that LSTM has obvious advantages. Kim et al. [24] found that LSTM can effectively improve the prediction performance of BDI by

comparing time series analysis and deep learning methods. In a study by Xiao et al. [25], they combined LSTM and integrated learning techniques to forecast CBCFI, which outperformed other methods when dealing with information involving dramatic market downturns. In stock market research, Nelson et al. [26] proposed a stock price prediction model based on LSTM, which was used to simulate transactions and compared with the benchmark model to evaluate its prediction performance. The experimental results showed that the LSTM-based model had lower risk compared with other models in trading simulation. Nabipour et al. [27] found that the LSTM model has stronger fitting ability than other machine learning models. Although the LSTM model exhibits strong predictive ability in time series, it can only obtain information from a single forward timeline during weight training. The BiLSTM model makes up for the shortcomings of the LSTM model. Many scholars have confirmed that BiLSTM has better performance than LSTM in the field of prediction [28, 29].

As shipping indices are a complex system influenced by multiple external factors and the shipping price series itself possesses a large amount of noise, more and more scholars recognize that a single neural network model alone cannot adequately extract the key features of a complex series. It has become a challenge to combine existing forecasting methods to improve forecasting effectiveness by decomposing time series data. Therefore, scholars have integrated machine learning models from disciplines such as financial econometrics and signal engineering in the hope of achieving accurate forecasting of financial time series, and analytical methods such as Fourier transform and wavelet transform have started to be applied in financial time series. Yang et al. [30] built a combined model of wavelet transform and support vector machine to predict BPI. The prediction results showed that the combined wavelet transform and SVM model had higher accuracy than the original single model. In addition, Han et al. [31] also tried to build a model combining wavelet transform and SVM to solve the prediction problem of shipping index. Leonov and Nikolov [32] used a hybrid model of wavelet and neural network to study the fluctuation of freight rates on the Baltic Panama 2A and Baltic Panama 3A routes. However, due to the difficulty of Fourier transform in handling mutant data, the non-adaptive drawbacks of wavelet analysis, and the high volatility and noise characteristics of shipping indices, the above two methods are not effective in dealing with this type of financial time series. Empirical modal decomposition techniques are usually applied to non-smooth and non-linear signals. Huang et al. [9], members of the US Academy of Engineering, proposed the Empirical Mode Decomposition (EMD) method, which can adaptively decompose a non-linear signal into multiple Intrinsic Mode Functions (IMFs) and can effectively suppress continuous noise such as Gaussian noise. Therefore, the combination of EMD and forecasting models is gradually being considered. Zeng et al. [20] used a combined EMD and ANN model to forecast the Baltic Dry Index. The research of Chen et al. [33] combined empirical mode decomposition (EMD), component reconstruction technology and grey wave prediction to simulate China Container Freight Index (CCFI). Using the EMD decomposition method, Li et al. [34] constructed a forecasting model with a combination of EMD decomposition GMDH of the AC algorithm for forecasting analysis of NYMEX crude oil futures, and the results showed that the forecasting model significantly improved the forecasting accuracy. Awajan et al. used EMD-HW bagging when predicting stock market data. The results show that the accuracy of the model is significantly improved after using EMD decomposition [35]. However, the EMD cannot suppress intermittent noise and mixed noise. To solve this problem, Wu et al. [36] proposed Ensemble Empirical Mode Decomposition (EEMD), which is an improved form of empirical mode decomposition, and the integrated empirical mode decomposition technique can solve the problem of noisy mode mixing. In the EEMD algorithm, a set of white noise is superimposed on the original signal and then decomposed into several IMFs. The average value of the corresponding IMF set is considered as the correct result and EEMD

will separate the noise in different IMFs from the original signal components, thus eliminating the noise mode mixing phenomenon. However, the EEMD approach requires a large number of averages to reduce the Gaussian white noise added during processing, which makes the computational time cost significantly higher. Yeh and Shieh [37] proposed the Complementary Ensemble Empirical Mode Decomposition (CEEMD), which uses a more complex method, adding a pair of opposite white noise signals to the original signal. And then performing the EMD decomposition separately, ensures that the decomposition is at least as good as EEMD, while the method reduces the reconstruction error caused by white noise and can significantly reduce the prediction difficulty of the model.

In general, most of the existing literature on shipping index forecasting models are traditional econometric models or simple machine learning models, and there are few papers that combine empirical modal decomposition with neural network models and apply them to the field of shipping market forecasting. Since deep learning has shown good prediction performance in various fields, we hope to explore the effectiveness of deep learning in shipping index prediction in this article. Based on the above literatures, this article proposes a combined model—CEEMD-PSO-BiLSTM model by using the idea of "decomposition-reconstruction-integration" and combining the empirical mode decomposition method with neural network model. The above-mentioned method firstly decomposes the shipping index by CEEMD, recombines the IMF by reconstruction, then constructs PSO-BiLSTM model prediction of each sub-sequence in turn, and finally integrates the prediction results to get the final prediction results. In this article, six major shipping indices (Far Eastern Dry Index, Baltic Dry Index, Sea Bulk Composite Freight Index, China Coastal Coal Freight Index, New York Crude Oil Index and China Import Crude Oil Freight Index) in the Chinese shipping market are selected as test objects, while existing mainstream machine learning forecasting models are compared, as well as different combination models based on them, to verify the effectiveness of the method in this article.

## 3. Methodology

### 3.1 CEEMD model

CEEMD is based on EMD. In order to solve the mode aliasing problem of EMD in IMF decomposition, several groups of independent and identically distributed white Gaussian noises are added to the original sequence, and then EMD decomposition is carried out to obtain IMF components, thus better solving the problem of the influence of EEMD on the accuracy of the original sequence. The specific CEEMD algorithm process is as follows.

1. Generate T groups of positive and negative pairs of Gaussian white noise data sets are added to the original sequence, so there are 2T signal sets:

$$\begin{bmatrix} x_1 \\ x_2 \end{bmatrix} = \begin{bmatrix} 1 & 1 \\ 1 & -1 \end{bmatrix} \begin{bmatrix} X \\ \varepsilon_0 \omega^i \end{bmatrix} \qquad (1)$$

$$i = 1, 2, \cdots, T$$

$x_1$, and $x_2$ are the signals after the addition of positive and negative paired white noise, respectively. $X$ is the original time series. $\omega^i$ is the Gaussian white noise subject to normal distribution $\varepsilon_0$ is the standard deviation of the added noise; $T$ is the number of times the noise is added (or the number of pooled samples).

2. Using EMD method to decompose the sequence data, each signal gets a group of IMF components. where the $j$th component in the $i$th decomposition is denoted as $imf_j^i$. The corresponding IMF components are integrated and averaged to obtain each IMF component $\widetilde{imf_j}$:

$$\widetilde{imf_j} = \frac{1}{2T} \sum_{i=1}^{2T} imf_J^i \tag{2}$$

3. The final decomposition result of CEEMD is:

$$X = \widetilde{imf_1} + \widetilde{imf_2} + \cdots + \widetilde{imf_j} + res = \sum_{k=1}^{j} imf_k + res \tag{3}$$

## 3.2 LSTM model

The LSTM network was first proposed and designed by Hochreiter, and then improved by Schmidhuber et al. [23]. As shown in Fig 1, it is proposed to add forgetting gates to the model to avoid gradient explosion or gradient disappearance in RNN, which is suitable for continuity prediction. Simple cyclic neural network consists of input layer, hidden layer and output layer.

Specifically, one neuron in LSTM model contains one cell and three gate mechanisms. Cell state is the key of LSTM model, which is similar to memory and is the memory space of the model. Cell state changes with time, and the recorded information is determined and updated by gate mechanism. The gate mechanism is a method to let information selection pass through, which is realized by sigmoid function and dot multiplication operation. Sigmoid takes a value between 0 and 1. And the multiplication, or dot product, determines the amount of information that is transferred (how much of each part can pass). when Sigmoid takes 0 it means that information is discarded, and when it takes 1 it means that it is fully transferred (i. e. fully remembered).

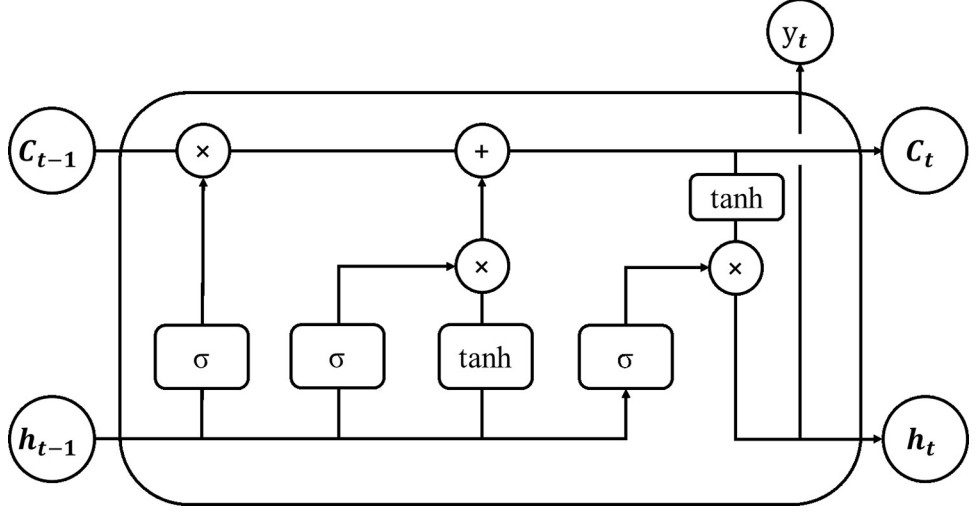

**Fig 1. Internal structure of neurons in LSTM model.**

The LSTM has three gates to protect and control cell state, which include a forget gate, an update gate and an output gate. Wherein the forgetting gate determines how many memory states are removed in the previous moment, that is, how many memories are left, and controls the input of the hidden state a (t-1) at the previous moment and the current moment X (t), and the activation of Sigmoid, which selectively removes the old information at the previous moment; The input gate determines how much new input information needs to be stored in the memory state at the current time, and controls the input of the hidden state a (t-1) at the previous time and the current time X (t), and the activation of Sigmoid, which determines how much new input information needs to be stored in the memory state at the current time; The output gate determines how many memory states are used for output at the current time, including the inputs of hidden state a (t-1) at the previous time and X (t) at the current time, and the activation of Tanh and Sigmoid. The information transmission is completed by the above three gates.

1. Forget gate

The network determines what information is forgotten from the cellular state through the Sigmoid function of the forgetting gate, with the following equation.

$$\Gamma_f = \sigma(w_f[a^{<t-1>}, x^{<t>}] + b_f \tag{4}$$

$a^{<t-1>}$ is the output representing the moment (t-1). $x^{<t>}$ is the input representing this layer at moment t. $w_f$ is the weight of each variable. $b_f$ is the representative bias term. $\sigma$ is the sigmoid function of the form $\sigma(x) = (1+e^{-x})^{-1}$; the $\Gamma_f$ represents the output to each of the values in the cell state $c^{<t-1>}$ in the cell state, between 0 and 1, where 1 means "keep all" and 0 means "discard all".

2. Update gate

The update gate is responsible for updating the information stored in the cell in a three-step operation.

Step 1: Update the results of the sigmoid function calculation for the gate $\Gamma_u$ to determine which values need to be updated.

Step 2: Create a new vector of candidate values based on the tanh function $\widetilde{c}^{<t>}$ and add it to the cell state.

Step 3: By multiplying the old cell state by the forgetting gate ($\Gamma_u$), like human memory, to forget some of the past information and subsequently add new information to the memory. The exact formula is shown below.

$$\Gamma_u = \sigma(w_u[a^{<t-1>}, x^{<t>}] + b_u \tag{5}$$

$$\widetilde{c}^{<t>} = tanh(w_c[a^{<t-1>}, x^{<t>}] + b_c) \tag{6}$$

$$c^{<t>} = \sigma\Gamma_u * \widetilde{c}^{<t>} + \Gamma_f * c^{<t-1>} \tag{7}$$

$tanh$ is a hyperbolic tangent function of the form $tanh(x) = \frac{sinh(x)}{cosh(x)} = \frac{e^x - e^{-x}}{e^x + e^{-x}}$; $c^{<t-1>}$ is the value of the state of the cell at moment t—1. $\widetilde{c}^{<t>}$ is the information to be remembered extracted from the input information at moment t. $\Gamma_u * \widetilde{c}^{<t>}$ is the new added value; and $c^{<t>}$ is the updated cell state value.

3. Output gate

The output gate determines the information that is output, which is based on the current cell state output. The sigmoid function is used to determine which part of the information is output, and the tanh function is used to process $c^{<t>}$, the $\Gamma_o$ and $c^{<t>}$ multiplying by each other to obtain the output value at moment t. The equation is shown below.

$$\Gamma_o = \sigma(w_o[a^{<t-1>}, x^{<t>}] + b_o \tag{8}$$

$$a^{<t>} = \Gamma_o * c^{<t>} \tag{9}$$

The internal processing of 1 neuron is accomplished through three gate mechanisms: forget gate, update gate and output gate. The LSTM model formed by multiple neurons in series has a selective memory function, allowing the model to form memories of long periods of past data.

### 3.3 BiLSTM model

The LSTM model is only a unidirectional neural network. A unidirectional network model can only receive information transmitted in the forward direction, and in practical applications, the output results may be influenced by the combination of the preceding and following information. In the case of time series prediction, a bi-directional LSTM model can effectively constrain the range of results of the training set operations and make the network structure more generalisable, thus optimising the fitting effect to the test set. As shown in Fig 2, the BiLSTM network structure consists of four layers: Input Layer, Forward Layer, Backward Layer and Output Layer.

As can be seen in the network structure in Fig 2, the Forward and Backward layers together influence the output layer, where $w_1-w_6$ are six shared weight values. The data is computed once in the Forward layer in the forward direction and the output of the hidden layer is saved

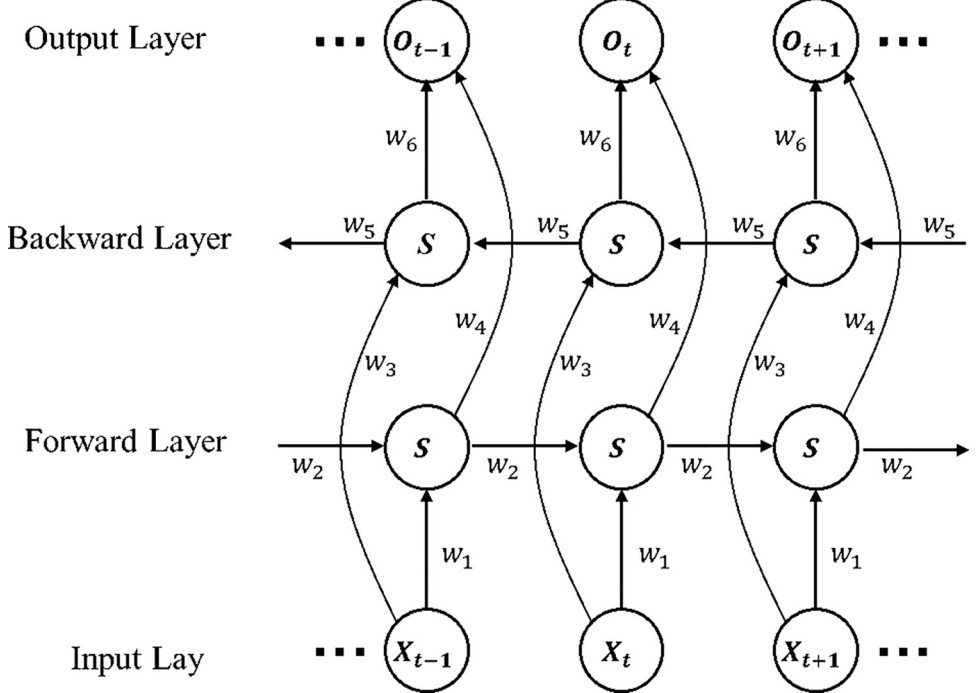

**Fig 2. BiLSTM network structure.**

for each moment forward. The data is computed once in the Backward layer in the reverse direction and the output of the backward hidden layer is stored at each moment. The final output is obtained by combining the results of the Forward and Backward layers at the corresponding moments. The mathematical expression of the process steps is as follows.

$$h_t = f((w_1 x_t + w_2 h_{t-1}) \tag{10}$$

$$h'_t = f((w_3 x_t + w_5 h'_{t-1}) \tag{11}$$

$$o_t = g((w_4 h_t + w_6 h'_t) \tag{12}$$

## 3.4 Particle swarm optimization algorithm

Particle Swarm Optimization (PSO) is an algorithm proposed by Eberhart and Kennedy in 1995 [38]. The algorithm was originally an optimization algorithm constructed by simulating the foraging behavior of a flock of birds. The idea of PSO itself is: particles constructed in the multi-dimensional space, each particle can be regarded as an individual bird foraging in the multi-dimensional space, and the location coordinates of food can be regarded as the parameters of the global optimal solution. The search for an optimal solution by a particle can be likened to the flight of a bird in search of food. The flying speed of particles can be dynamically adjusted according to the historical optimal position of particles and the historical optimal position of population. A particle has only two properties, speed and position, speed represents the speed of movement, position represents the direction of movement. The optimal solution searched individually by each particle is called the individual extremum. The optimal individual extremum in the particle swarm is taken as the current global optimal solution. In the iteration process, the global optimal solution is finally obtained by updating the filling speed and position. The basic formula of PSO is as follows:

$$V_{i,j}^{t+1} = \omega V_{i,j}^t + c_1 r_{1,i,j}^t (\widehat{y}_{i,j}^t - x_{i,j}^t) + c_2 r_{2,i,j}^t (y_{i,j}^t - x_{i,j}^t) \tag{13}$$

$$x_{i,j}^{t+1} = x_{i,j}^t + V_{i,j}^{t+1} \tag{14}$$

t is the number of iterations, i is the ith particle, and j is the jth dimension; $V_{i,j}^t$ is the velocity of i particles in the jth dimension at the t th iteration; $\omega$ is the inertia weight; c1 and c2 represent the two acceleration coefficients; $x_{i,j}^t$ is the spatial position of i particle at t iterations, $y_{i,j}^t$ is the space extreme point of t iterations; $r_{1,i,j}^t$ and $r_{2,i,j}^t$ are random numbers with uniform distribution in [0,1]. According to the formula, it can be observed that the velocity of the ith particle in the jth dimensional space at time t+1 is determined by three regions: the first is the velocity of the particle at time t, representing the inertia of the particle; The second is the influence of the previous trajectory of the particle in space on the direction of the subsequent movement; The third is the influence of the trajectories of all particles in space on the direction of each particle's subsequent motion.

It can be divided into the following 6 steps:

1. Initialize parameters. It includes setting the upper and lower limits in the data space, two acceleration coefficients c1 and c2, the maximum iteration coefficient max_ episode, the maximum and minimum velocity of each particle. The initial position and velocity of each particle are randomly set.

2. Define the fitness function, calculate the fitness at the initial position of each particle, save the fitness at the initial global optimum and the current spatial position and velocity of each particle.

3. Update the velocity and position of each particle under the current iteration number according to Eq (13) and Eq (14).

4. Determine the fitness of each particle after moving once according to the fitness function, and compare the current optimal fitness with the historical individual optimal value. If the current fitness is superior, the historical individual optimal fitness is replaced by the current optimal fitness, and the current particle is updated at the same time. If the historical global optimum is superior, the replacement process is not performed.

5. Determine the global optimal fitness of each individual particle after updating. If the global optimum fitness is better than the initial global optimum, the global optimum is updated.

6. Determine whether the iteration cycle meets the stopping condition (the stopping condition can be set to meet the accuracy requirements of the experimental purpose or to reach the maximum number of iterations). If the conditions are met, the current optimal value and optimal parameters are output. If the condition is not met, repeat Step 3 to Step 6 until the condition is met.

## 3.5 Model construction of CEEMD-PSO-BiLSTM for shipping index

According to the above, the shipping index has the characteristics of non-stationary, non-linear and high complexity, and a single deep learning prediction method cannot accurately grasp its main characteristics for prediction. The specific modeling process of CEEMD-PSO--BiLSTM model is shown in Fig 3. Using the idea of "decomposition-reconstruction-integration", firstly, the original time series can be decomposed into multiple eigenmode components by CEEMD decomposition method. Different eigenmode function components reveal the characteristics of shipping index in different time scales, and the high-frequency components and low-frequency components are distinguished by single sample T test. Then the low-frequency components are reorganized; The reorganized sequences are predicted by deep learning model respectively, and combined with the excellent performance of BiLSTM in long memory in time series prediction, each component is characterized, learned and fitted to achieve accurate prediction of each component; Finally, the integrated method is used to recombine the prediction results of each component to form the final prediction results. The specific modeling steps are as follows:

Step 1: Carry out CEEMD decomposition on the original sequence of shipping index to obtain n intrinsic modal function components (IMF in turn)$_1$, $IMF_2 IMF_N$) and the trend item RES.

Step 2: Pass a one-sample t-test with zero mean for all eigenmodal function components, and identify the first component with $\alpha > 0.05$ (set as $IMF_m$) and its subsequent components as the low-frequency component ($IMF_m, \ldots, IMF_n$), which reflects the cyclical trend of the shipping index, and combine the low-frequency components into a new component LF; The previous components of IMF are the high frequency components ($IMF_1, IMF_2, \ldots, IMF_{m-1}$), which indicate random fluctuations in the short term; RES is the trend term, which reflects the long-term trend of the original series.

Step 3: The $IMF_1, IMF_2, \ldots, IMF_{m-1}$, LF, and RES subsequences are modeled and predicted using BiLSTM neural networks, respectively. The predicted results are $\widehat{IMF}_1, \widehat{IMF}_2, \ldots, \widehat{IMF}_{m-1}, \widehat{LF}$ and $\widehat{RES}$, respectively, and PSO algorithm was used to optimize the hyperparameters of each prediction.

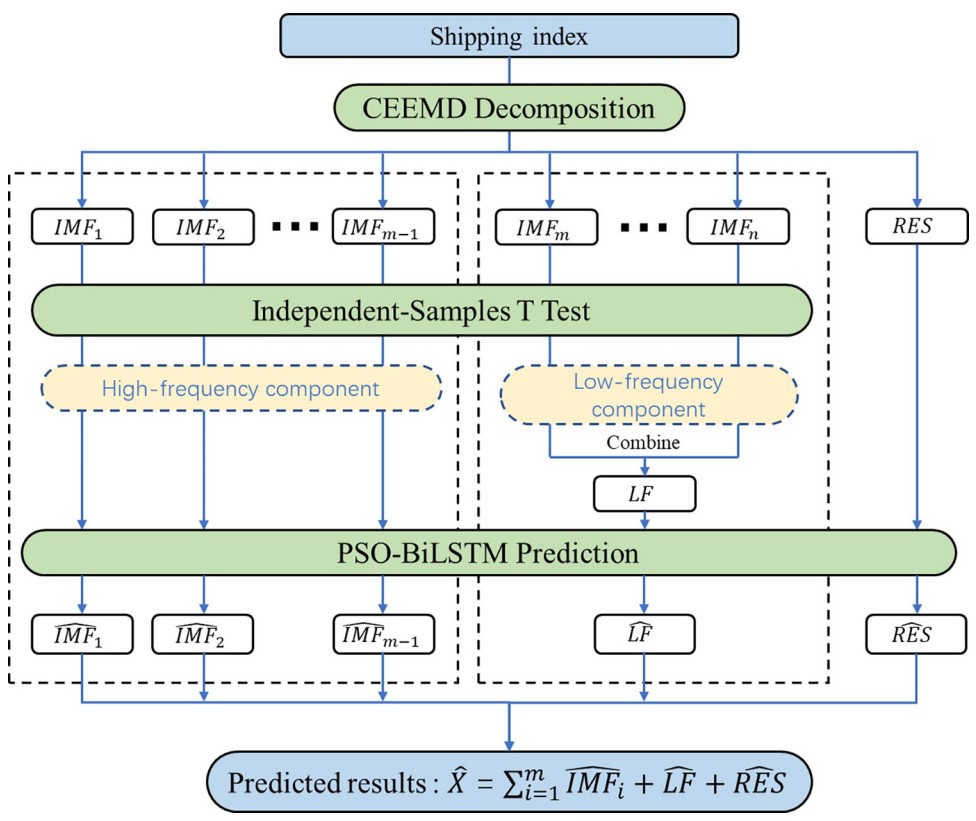

**Fig 3. CEEMD-PSO-BiLSTM network structure.**

Step 4: Combine the predictions from step 3. The prediction results of the original sequence are obtained by integration processing.

# 4. Empirical analysis

## 4.1 Data sources

This article focuses on six major Chinese shipping indices, including the price series along the Baltic Dry Index (BDI), the Consolidated Maritime Bulk Freight Index (CBFI), the China Coastal Coal Freight Index (CBCFI), the New York Crude Oil (CFD), the Far East Dry Bulk Composite Index (FDI) and the China Crude Oil Import Freight Index (CTFI), with data from the RESSET database. Table 1 shows the descriptive statistics of the six shipping indices, with the start date and end date of the data selected in this article, where CBFI is measured in weeks

**Table 1. Descriptive statistics of six shipping indexes.**

| Shipping index | Training set | Test set | unit | N | Minimum | Maximum | Mean | Std. | Skewness | Kurtosis |
|---|---|---|---|---|---|---|---|---|---|---|
| CBFI | 2004/1/2-2016/09/22 | 2016/09/23-2021/10/15 | week | 868 | 771.01 | 2886.92 | 1294.08 | 381.206 | 1.398 | 2.323 |
| CBCFI | 2011/12/7-2018/11/14 | 2018/11/15-2021/10/21 | day | 2330 | 370.99 | 1706.2 | 759.58 | 244.298 | 0.985 | 0.644 |
| BDI | 2000/1/4-2015/04/21 | 2015/04/22-2021/10/25 | day | 5477 | 290 | 11793 | 2292.24 | 2006.639 | 2.118 | 4.889 |
| CFD | 2000/1/4-2015/03/16 | 2015/03/11-2021/10/26 | day | 5585 | 11.57 | 145.29 | 61.13 | 25.866 | 0.429 | -0.607 |
| FDI | 2017/11/28-2020/09/01 | 2020/09/05-2021/10/26 | day | 928 | 471.54 | 3199.55 | 1159.32 | 530.45 | 1.664 | 2.451 |
| CTFI | 2013/11/28-2019/06/12 | 2019/06/13-2021/10/26 | day | 1903 | 482.63 | 4925.76 | 946.35 | 461.769 | 2.888 | 13.716 |

and the remaining five shipping indices are measured in days. The skewness and kurtosis values of the six shipping indices reflect the non-normal distribution of the shipping indices.

## 4.2. Shipping index decomposition based on CEEMD

Due to the similarity of the six shipping indices construction methods, the Far East Dry Bulk Composite Index (FDI) is used here as a representative for analysis and research. Both the EMD and CEEMD operations taken in this experiment were performed in Matlab 2020a. In this article, the CEEMD decomposition integration number NE is set to 1000, the ratio of additional noise standard deviation to the original series standard deviation is 0.2, and the maximum number of iterations for each component is 1000. The CEEMD decomposition results are shown in Fig 4.

As can be seen from the left of Fig 4, the first series is the raw data of the Far Eastern Dry Bulk Composite Index trading price from 28 November 2017 to 27 October 2021, the data is divided into 9 series after CEEMD decomposition, the first 8 series are the Intrinsic Modal Fraction (IMF), from the figure we can find that IMF1~IMF8 in order down its fluctuation frequency gradually decreases, the last one is the res. The first part of the IMF component series shows a complex irregular and high frequency fluctuation image, and its mean value fluctuates above and below 0. It is called high frequency component, and the high frequency component can be regarded as the short-term fluctuation of the Far East Dry Index time series, such as short-term investor sentiment, short-term policy fluctuations and other factors, these factors cause the shipping market price fluctuations in the short term; The IMF series more backward have the lower frequency of fluctuation, which are called the low frequency component,

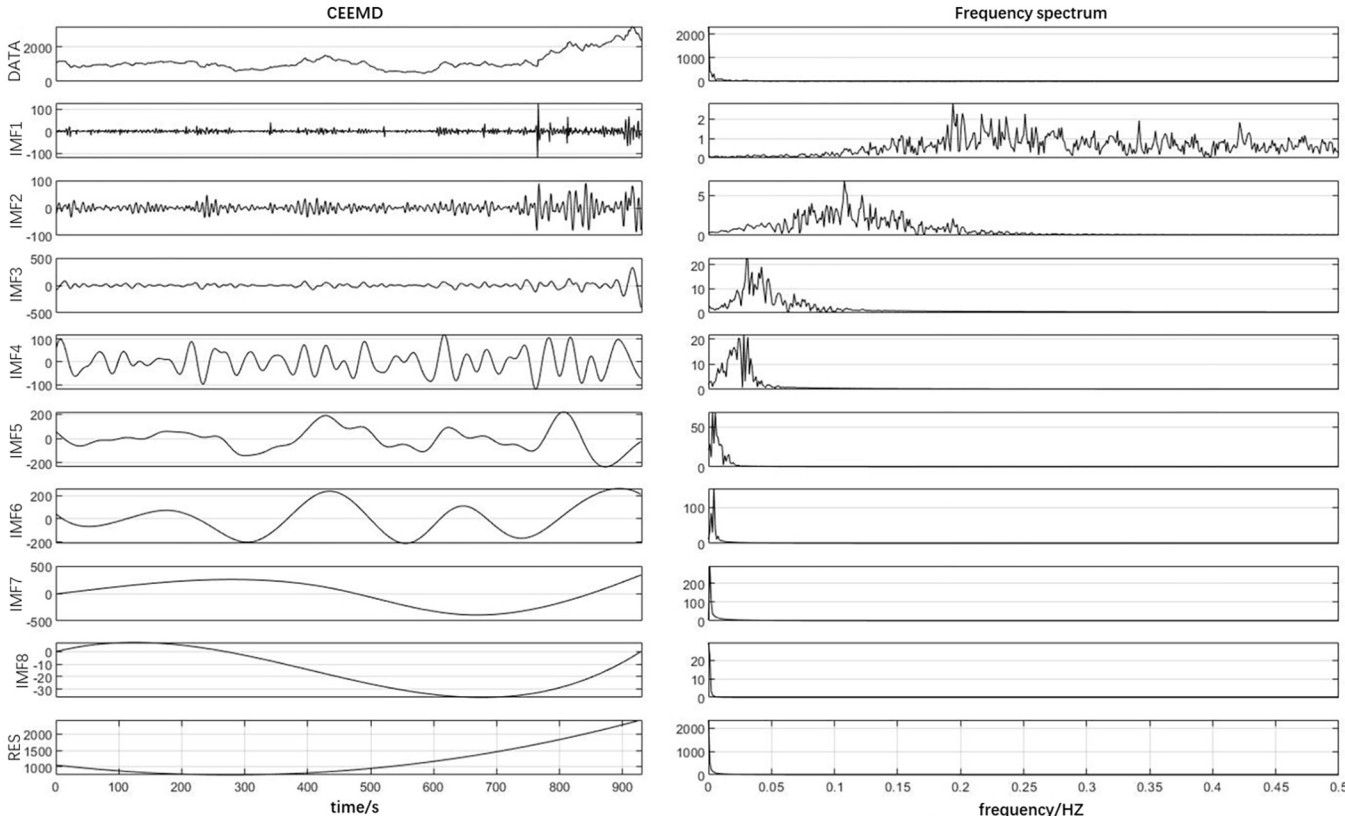

**Fig 4. Sequences and spectra of FDI components after CEEMD decomposition.**

**Table 2. Single sample T test based on CEEMD decomposition.**

| Component | Z Z = 0 | | | | | |
|---|---|---|---|---|---|---|
| | **T-value** | **df** | **Sig. (2-tailed)** | **Mean difference** | **Difference 95% confidence interval** | |
| | | | | | **Lower Bound** | **Upper Bound** |
| IMF1 | -0.106 | 928 | 0.916 | -0.041 | -0.791 | 0.710 |
| IMF2 | 0.237 | 928 | 0.812 | 0.149 | -1.081 | 1.379 |
| IMF3 | 0.917 | 928 | 0.360 | 1.543 | -1.761 | 4.846 |
| IMF4 | 0.624 | 928 | 0.532 | 0.992 | -2.126 | 4.111 |
| IMF5 | -3.461 | 928 | 0.001 | -10.475 | -16.414 | -4.536 |
| IMF6 | 1.233 | 928 | 0.218 | 5.396 | -3.191 | 13.983 |
| IMF7 | -0.306 | 928 | 0.759 | -2.255 | -16.699 | 12.189 |
| IMF8 | -28.510 | 928 | 0 | -14.744 | -15.759 | -13.729 |
| res | 76.278 | 928 | 0 | 1178.752 | 1148.424 | 1209.079 |

showing the occurrence of some major events in the shipping market, the implementation of important policies or the results of the impact of economic cycles, reflecting the long-term change pattern of the time series; the last item is called the trend term series, and the trend of the res component turning from smooth to upward can be seen in Fig 4, reflecting the long-term fluctuation of the Far Eastern Dry Bulk Composite Index steadily increasing in four years trend.

This article further plots the spectrum of each component based on the Fourier Transform (FFT) measure, thus providing a more intuitive analysis of the impact of each IMF on the FDI price series. The horizontal axis of the spectrum is frequency and the vertical axis is amplitude, where the lower the frequency of a component, the more profound the influence of that component on the original series, and the larger the amplitude, the greater the influence of that component on the original series. The spectrum in the right of Fig 4 shows that the frequency of IMF1 to IMF8 decreases and its amplitude increases, indicating that the lower frequency component has a more significant and far-reaching effect than the higher frequency component, which is characterized by periodic fluctuations.

To determine the high-frequency and low-frequency components of the IMF series, the method used by Li and Feng [39] is referenced here, where the high-frequency component is assumed to fluctuate up and down around a mean of 0. A one-sample t-test with a mean of 0 is conducted for each IMF, and the first IMF series that deviates from a mean of 0 is used as a marker to distinguish the high-frequency component from the low-frequency component.

From Table 2, we can learn that the one-sample t-test for the IMF1 sequence has a t-value of -0.106, a mean difference of -0.041, and a significance p-value of 0.916 greater than 0.05 indicating that the mean of this sequence is not different from zero at the $\alpha = 0.05$ The IMF5 sequence is judged to be a low frequency component as its p-value of 0.001 is less than 0.05, indicating that the mean of IMF5 is significantly different from zero. In particular, we can see from Table 2 that the p-values of IMF6 and IMF7 sequences are 0.218 and 0.759 respectively, which are greater than the given significance level of 0.05, indicating that their mean values are not significantly different from zero, but we still judge them as low-frequency components because IMF4 was judged to be a low-frequency component in the previous step, and the frequency of IMF sequences is gradually decreasing as shown in the spectrum of Fig 4, so even though IMF4 was still judged to be a low-frequency component in the previous step.

## 4.3 Selection of evaluation indicators

The experiments in this article were done on python 3.7, built and run by Keras+TensorFlow 2.5. The FDI decomposed and reconstructed sequences for high- The FDI decomposed and

reconstructed sequences for high- frequency component, low-frequency component, and trend term res. BiLSTM prediction models were constructed for each subsequence separately, and then the models were used to make rolling predictions for each subsequence in the prediction interval with root mean square error (RMSE), mean absolute percentage error (MAPE) percentage error (MAPE), mean absolute error (MAE) and coefficient of determination ($R^2$) four evaluation metrics.

1. RMSE

The range is [0, +∞), when the predicted value and the true value are completely equal to 0, it is a perfect model. the greater the error, the greater the value.

$$RMSE = \sqrt{\frac{\sum_{i=1}^{n} (\widehat{y}_i - y_i)^2}{n}} \tag{15}$$

2. MAE

The range is [0, +∞). When the predicted value is exactly equal to the true value, the MAE is equal to 0, which is a perfect model. the greater the error, the greater the MAE.

$$MAE = \frac{1}{n} \sum_{i=1}^{n} |\widehat{y}_i - y_i| \tag{16}$$

3. SMAPE

The range is [0, +∞). If the value of SMAPE is 0, it is a perfect model, and if the value of SMAPE is greater than 100%, it means an inferior model.

$$MAPE = \frac{100\%}{n} \sum_{i=1}^{n} \frac{|\widehat{y}_i - y_i|}{|\widehat{y}_i| + |y_i|/2} \tag{17}$$

4. $R^2$

The range is [0, 1], $R^2$ reflects the extent to which the independent variable x explains the changes in the dependent variable y. The closer its value is to 1, the better the model fits.

$$R^2 = 1 - \frac{\sum_{i=1}^{n} (\widehat{y}_i - y_i)^2}{\sum_{i=1}^{n} (\bar{y}_i - y_i)^2} \tag{18}$$

Where $y_i$ is the true value of the series, $\widehat{y}_i$ is the sequence predicted value, $\bar{y}_i$ is the sequence mean, and n is the total number of predicted data.

In order to eliminate the influence of different dimensions between different sequences and improve the operational efficiency of the model, Z-score is used to standardize the data of each sequence before predicting the model. The first 70% data of the sample is set as the training set, the verification set is the first 10% of the training set, and the last 30% data of the sample is the test set. The rolling prediction modeling method is adopted to predict the price of Far East Dry Bulk Composite Index in the next period through the transaction data of the first 10 days, and the look_back is set as the data amount within 10 days. The standardized series are predicted by BiLSTM model respectively. Taking FDI as an example, after repeated adjustment

and omparison of parameters, the first layer is LSTM layer with 32 nodes, the second layer is BiLSTM layer with 32 nodes, the number of training samples batch_size is set to 6, the iteration times epoch is set to 50, and the optimizer is selected as Adam.

## 4.4 Reorganization and optimization of IMF sequence

The analysis in the previous section shows that the high frequency component contains a lot of noise and short-term random influences, while the low frequency component is a smoother time series. If the high-frequency components are combined for prediction, the large amount of noise in the high-frequency components will not cancel each other out, but will instead amplify the effect of their random factors, so the high-frequency components can be predicted one by one; while the low-frequency components themselves are smoother, the low-frequency components are combined into a new series (named LF) for prediction, in order to improve the prediction efficiency of the model. The specific formula is as follows.

$$CEEMD-BiLSTM = \sum_{i=1}^{m} \widehat{IMF_i} + \widehat{LF} + \widehat{res} \tag{19}$$

In here, m denotes the number of high-frequency components, and $\widehat{IMF_i}$ denotes the predicted value of each HF component, and $\widehat{LF}$ denotes the predicted value of the LF after recombination of the low frequency components, and $\widehat{res}$ denotes the predicted value of the trend term res.

In this article, we take the FDI as an example, and the reconstructed components are shown in Fig 5. The process of processing the data of the other five shipping indices is similar to that of FDI, which will not be explained in detail due to space limitation. The three high-frequency components IMF1, IMF2, IMF3 and IMF4 of the FDI after CEEMD decomposition, the new series LF and trend term res after the reorganization of the low-frequency components are predicted by the BiLSTM model, and the prediction results of each series are added together to form the final prediction results, and the combined prediction method is named CEEMD--BiLSTM1:

$$CEEMD-BiLSTM1 = \widehat{IMF}_1 + \widehat{IMF}_2 + \widehat{IMF}_3 + \widehat{IMF}_4 + \widehat{LF} + \widehat{res}$$

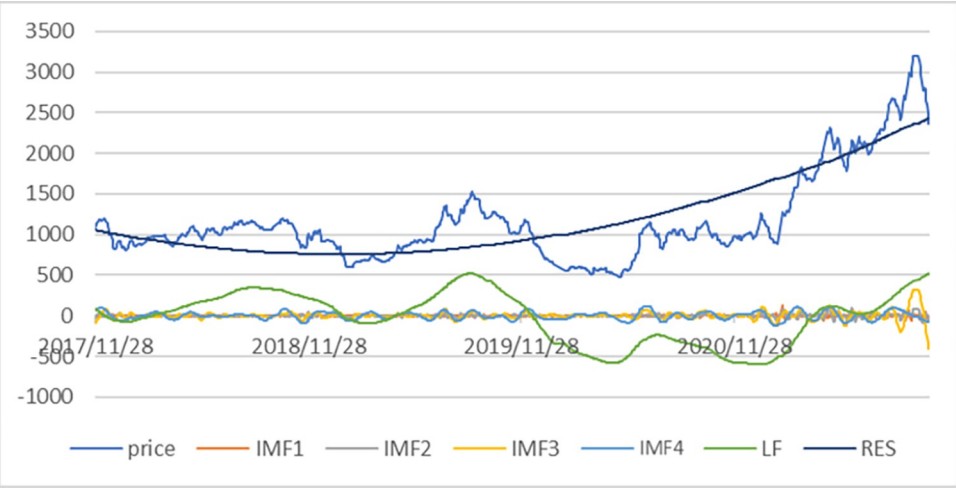

**Fig 5. CEEMD decomposition and reorganization diagram of FDI.**

In order to verify the superiority of the IMF recombination method in this article, the high frequency components are recombined according to the previous practice, and the recombined new sequence is named HF, HF and LF, and the trend term res are predicted and summed by BiLSTM model, this method is named CEEMD-BiLSTM2; the HF and 4segment low frequency components IMF5 to IMF8, and res are summed up by BiLSTM model prediction, and the combined prediction method is named CEEMD-BiLSTM3; the decomposed components are predicted directly by BiLSTM model without any processing, and this method is named CEEMD-BiLSTM4:

$$CEEMD-BiLSTM2 = \widehat{HF} + \widehat{LF} + \widehat{res}$$

$$CEEMD-BiLSTM3 = \widehat{HF} + \sum_{i=4}^{8} \widehat{IMF}_i + \widehat{res}$$

$$CEEMD-BiLSTM4 = \sum_{i=1}^{8} \widehat{IMF}_i + \widehat{res}$$

Using the model parameters determined in this article, the optimal HF IMF was determined based on the results of the prediction evaluation index, as shown in Table 3, after BTC was processed with 3 different IMF combination methods after CEEMD decomposition and predicted by the BiLSTM model, it can be seen that the first combination method is the optimal combination, and its RMSE, SMAPE, MAE, and $R^2$ 21.45%, 24.23%, 31.63% and 0.4% respectively over the second combination method, 58.61%, 64.25%, 65.23% and 2.95% respectively over the third combination method, and 53.4%, 62.1%, 63.55% and 2.18% respectively over the fourth combination method. It can be seen that the first combination approach chosen in this article is advantageous in forecasting the Far East Dry Bulk Composite Index. Fig 6 shows the forecasting effect under the four IMF combination methods.

## 4.5 PSO to optimize the BiLSTM

According to the experiments in the previous section, we will adopt the combination of CEEMD-BiLSTM1 for prediction. However, since the complexity of each sequence after CEEMD decomposition and recombination is different, it is unscientific to use the same hyperparameters in the model. Therefore, in order to match the model network structure with the characteristics of shipping index, we used PSO algorithm to optimize the model hyperparameters separately when each sequence was predicted by the BiLSTM model.

We set the number of neurons in LSTM layer and BiLSTM layer and the size of batch size as the optimization object of PSO algorithm, so the dimension is 3, the range of LSTM layer is 16~64, the range of BiLSTM layer is 16~64, and the range of batch size is 6~16. Set the parameter ω to be linearly decreasing from 0.9 to 0.4, c1 to be linearly decreasing from 2.5 to 0.5, c2 to

**Table 3. Evaluation results of FDI under four IMF portfolio modes.**

| Different combination models | RMSE | MAE | SMAPE | $R^2$ |
|---|---|---|---|---|
| CEEMD-BiLSTM1 | 51.219528 | 34.090326 | 1.773876 | 0.994114 |
| CEEMD-BiLSTM2 | 65.20678 | 44.989458 | 2.59466 | 0.990461 |
| CEEMD-BiLSTM3 | 123.742176 | 95.365907 | 5.101343 | 0.965646 |
| CEEMD-BiLSTM4 | 109.924022 | 89.74186 | 4.866178 | 0.97289 |

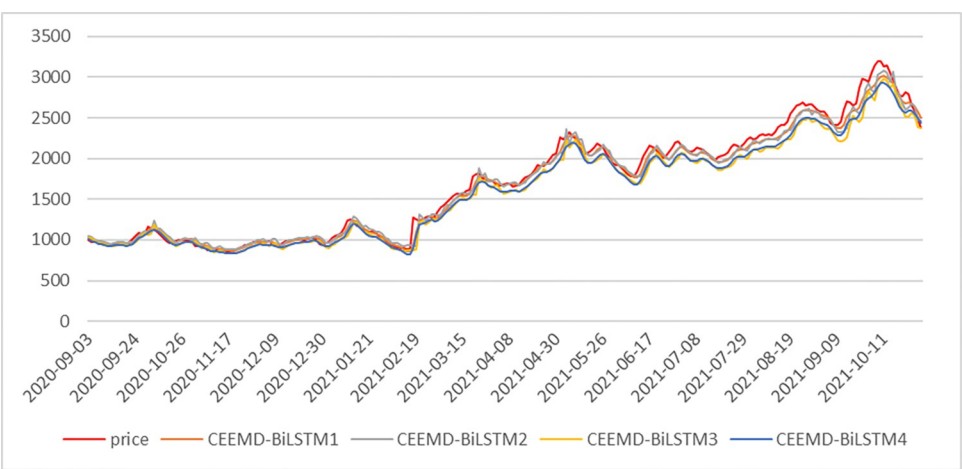

**Fig 6. Comparison of forecasting effects of FDI under four IMF combinations.**

be linearly increasing from 0.5 to 2.5, and the number of particles is 10; In order to avoid PSO falling into local optimum, we dynamically adjust the acceleration according to different iteration times. Acceleration as a function is:

$$c = c * (episode_i / episode_{max}) \qquad (20)$$

$c$ is the acceleration c1 and c2, $episode_i$ is the current iteration number, $episode_{max}$ is the maximum iteration number. We set the maximum iteration number in this experiment as 20.

The function of particle fitness is the average percentage error:

$$f = \frac{1}{N} \sum_{i=1}^{K} \frac{|\widehat{y}_i - y_i|}{y_i} \qquad (21)$$

N is the length of test set data; $\widehat{y}_i$ is the predicted value, and $y_i$ is the actual value.

As can be seen from Table 4, the accuracy of the four indexes has been significantly improved after PSO optimization. CEEMD-PSO-BiLSTM is the optimal combination. Compared with CEEMD-BiLSTM, the RMSE, SMAPE and MAE of CEEMD-BILSTM are reduced by 15.12%, 13.77% and 7.44%, respectively, and the R^2 is increased by 0.16%. It can be seen that the PSO algorithm can better improve the accuracy of the model.

## 5. Results and discussion

As the construction process of CBFI, CBCFI, CFD, BDI and CTFI prediction models is similar to that of FDI, it is not repeated due to space limitation. In this article, we constructed Support Vector Regression (SVR), Recurrent Neural Network (RNN), Gate Recurrent Unit (GRU), Long Short Term Memory (LSTM), Bi-directional Long Short-Term Memory (BiLSTM) as a comparison to verify the advantages of the BiLSTM model; meanwhile, to verify the effectiveness of the CEEMD decomposition method in shipping data, this article also decomposes the

**Table 4. FDI evaluation results before and after PSO optimization.**

|  | RMSE | MAE | SMAPE | R2 |
|---|---|---|---|---|
| CEEMD-BiLSTM | 51.2195 | 34.0903 | 1.7739 | 0.994114 |
| CEEMD-PSO-BiLSTM | 43.4701 | 29.3951 | 1.6419 | 0.99576 |

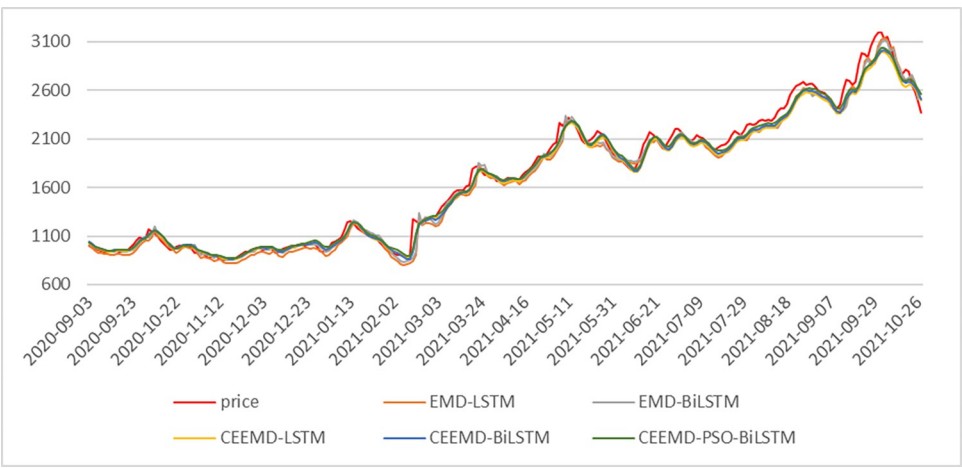

**Fig 7. FDI prediction results based on five combined models.**

original sequences based on the empirical modal decomposition (EMD) method, and the decomposition integration number NE is uniformly set to 1000, and the ratio of additional noise standard deviation to five combined models (CEEMD-PSO-BiLSTM, CEEMD-LSTM, CEEMD-BiLSTM, EMD-LSTM, EMD-BiLSTM) including the model in this article were constructed, and through the above ten models, it is hoped that the advantages of the CEEMD-PSO-BiLSTM model constructed in this article can be demonstrated. Among them, Fig 7 shows the comparison of the prediction results of the five combined models.

As can be seen from Table 5, the analysis and prediction of the six shipping indices by the "decomposition-reconstruction-integration" approach is significantly better than the prediction effect of a single model. After CEEMD decomposition, the prediction accuracy of the LSTM model and BiLSTM model was greatly improved. The CEEMD-PSO-BiLSTM model has the best results among the four evaluation indicators of the prediction results of the six shipping indices by the ten methods. In the time series model with more data, the prediction advantage of LSTM model in the long time series model can be given full play, when the prediction performance of LSTM model and BiLSTM model are similar, in CFD for example, BiLSTM model has less improvement in prediction effect than LSTM, in RMSE, MAE, SMAPE, $R^2$ compared to LSTM model respectively 2.19%, 2.03%, 2.47%, and 0.06%, which is due to the fact that the LSTM forward layer plays a major role in the same time series forecasting model, while the reverse layer can still capture information that may be overlooked in the forward layer, thus improving the forecasting performance of the model. In time series models with less data volume, the forecasting advantages of LSTM models in long time series models cannot be fully exploited, while BiLSTM neural network models obtained better forecasting results than LSTM after adding the inverse layer. Taking FDI as an example, in a single model, due to the small amount of FDI data, where in RMSE, MAE, SMAPE and $R^2$ are improved by 21.40%, 30.31%, 26.58% and 0.62% respectively compared to the LSTM model.

In the combinatorial model before optimization, the CEEMD-BiLSTM model, which is first decomposed by CEEMD and optimised and restructured, performs best, with RMSE, MAPE, MAE and $R^2$ improved by 23.02%, 15.75%, 22.87% and 0.41%, 20.02%, 34.62%, 47.96%, 0.33%, respectively, over the EMD-LSTM model, 8.63%, 10.91%, 15.72%, 0.12%, respectively, over the EMD-BiLSTM model, and 19.28%, 20.51%, 15.98%, 0.32%, respectively, over the CEEMD--BiLSTM model. 15.98%, and 0.32% respectively; After PSO optimization, CEEMD-PSO--BiLSTM model has good prediction accuracy in each shipping index. Fig 8 shows the SMAPE

**Table 5. Comparison of external prediction effects of six shipping indexes based on each model.**

| | | FDI | | | | CBFI | | | |
|---|---|---|---|---|---|---|---|---|---|
| | Evaluation Index | RMSE | MAE | SMAPE | $R^2$ | RMSE | MAE | SMAPE | $R^2$ |
| Single Model | LSTM | 84.6547 | 58.0644 | 3.1323 | 0.9839 | 55.5116 | 42.3965 | 3.6582 | 0.8437 |
| | BiLSTM | 66.5371 | 40.4645 | 2.2999 | 0.9901 | 47.5667 | 34.4102 | 3.0036 | 0.8852 |
| | SVR | 81.3575 | 59.3579 | 4.8695 | 0.9842 | 58.3205 | 43.4204 | 5.5711 | 0.7613 |
| | RNN | 75.7349 | 54.1851 | 2.9709 | 0.9871 | 50.6656 | 38.1451 | 3.3147 | 0.8721 |
| | GRU | 77.6491 | 57.3994 | 3.5249 | 0.9865 | 49.6632 | 36.8337 | 3.2458 | 0.8749 |
| Combination Model | EMD-LSTM | 64.0375 | 52.1442 | 3.4089 | 0.9908 | 37.9840 | 28.1666 | 2.4955 | 0.9268 |
| | EMD-BiLSTM | 56.0591 | 38.2640 | 2.1049 | 0.9929 | 34.9590 | 26.0642 | 2.2766 | 0.9380 |
| | CEEMD-LSTM | 63.4543 | 42.8890 | 2.1113 | 0.9910 | 16.8991 | 13.0838 | 1.1551 | 0.9855 |
| | CEEMD-BiLSTM | 51.2195 | 34.0903 | 1.7739 | 0.9941 | 14.5158 | 10.4990 | 0.9090 | 0.9893 |
| | CEEMD-PSO-BiLSTM | 43.4701 | 29.3951 | 1.6419 | 0.9958 | 13.9619 | 10.1241 | 0.8846 | 0.9901 |
| | | CBCFI | | | | BDI | | | |
| | Evaluation Index | RMSE | MAE | SMAPE | $R^2$ | RMSE | MAE | SMAPE | $R^2$ |
| Single Model | LSTM | 30.9609 | 16.2860 | 1.8322 | 0.9835 | 35.6277 | 23.3173 | 1.9257 | 0.9982 |
| | BiLSTM | 29.0716 | 14.0380 | 1.6336 | 0.9854 | 34.6795 | 21.8410 | 1.7501 | 0.9983 |
| | SVR | 30.0632 | 16.9301 | 4.0405 | 0.9841 | 103.3021 | 29.8333 | 3.5860 | 0.9835 |
| | RNN | 31.6914 | 18.5339 | 2.0094 | 0.9828 | 37.4036 | 24.0517 | 1.8626 | 0.9981 |
| | GRU | 29.6728 | 18.2126 | 2.1327 | 0.9848 | 36.5021 | 26.7041 | 2.4113 | 0.9981 |
| Combination Model | EMD-LSTM | 26.4743 | 14.4846 | 1.7710 | 0.9879 | 28.5650 | 19.9663 | 1.7064 | 0.9989 |
| | EMD-BiLSTM | 23.2467 | 12.3823 | 1.4981 | 0.9907 | 26.4251 | 16.0001 | 1.2329 | 0.9990 |
| | CEEMD-LSTM | 18.3232 | 13.8704 | 1.6398 | 0.9942 | 11.9949 | 9.2742 | 0.9435 | 0.9998 |
| | CEEMD-BiLSTM | 15.1484 | 11.4466 | 1.3878 | 0.9960 | 11.4739 | 9.0252 | 0.8716 | 0.9998 |
| | CEEMD-PSO-BiLSTM | 12.6338 | 8.1947 | 0.9157 | 0.9972 | 9.5114 | 6.1946 | 0.4928 | 0.9999 |
| | | CFD | | | | CTFI | | | |
| | Evaluation Index | RMSE | MAE | SMAPE | $R^2$ | RMSE | MAE | SMAPE | $R^2$ |
| Single Model | LSTM | 1.3955 | 0.9943 | 2.1340 | 0.9858 | 145.1201 | 64.0942 | 4.4978 | 0.9597 |
| | BiLSTM | 1.3650 | 0.9742 | 2.0812 | 0.9864 | 143.0641 | 48.9135 | 3.2929 | 0.9602 |
| | SVR | 2.5960 | 1.2580 | 3.6646 | 0.9436 | 285.8751 | 84.9773 | 6.7383 | 0.7335 |
| | RNN | 1.4509 | 1.0949 | 2.2868 | 0.9847 | 156.0857 | 55.7168 | 3.9140 | 0.9525 |
| | GRU | 1.5435 | 1.1326 | 2.3645 | 0.9826 | 147.2248 | 54.3780 | 4.1312 | 0.9578 |
| Combination Model | EMD-LSTM | 1.0013 | 0.7338 | 1.5456 | 0.9927 | 143.9901 | 58.0416 | 3.9877 | 0.9597 |
| | EMD-BiLSTM | 0.9435 | 0.6935 | 1.4702 | 0.9935 | 140.5251 | 56.0092 | 3.9102 | 0.9616 |
| | CEEMD-LSTM | 0.4851 | 0.3663 | 0.7838 | 0.9983 | 130.2206 | 49.5976 | 3.2489 | 0.9670 |
| | CEEMD-BiLSTM | 0.4763 | 0.3026 | 0.6938 | 0.9983 | 126.1977 | 47.4667 | 2.9995 | 0.9690 |
| | CEEMD-PSO-BiLSTM | 0.4388 | 0.3084 | 0.6694 | 0.9986 | 123.0814 | 47.6599 | 2.9946 | 0.9705 |

metrics for the six shipping indices in five different hybrid models, and it can be seen that the CEEMD-PSO-BiLSTM model has good prediction results in both long time series and short time series.

Analyzing the prediction performance of the above models in the six shipping indices, it can be found that: 1. the prediction effect of the integrated model constructed by the "decomposition-reconstruction-integration" method is significantly better than that of the single model; 2. the decomposition result of the CEEMD method for shipping data is better than that of the EMD decomposition method, indicating that CEEMD is better in feature decomposition; 3. The BiLSTM model can obtain better prediction results compared with the LSTM model, and its prediction effect is better in short time series. 4. The model optimized by PSO has better fitting effect and prediction accuracy. In summary, the CEEMD-PSO-BiLSTM

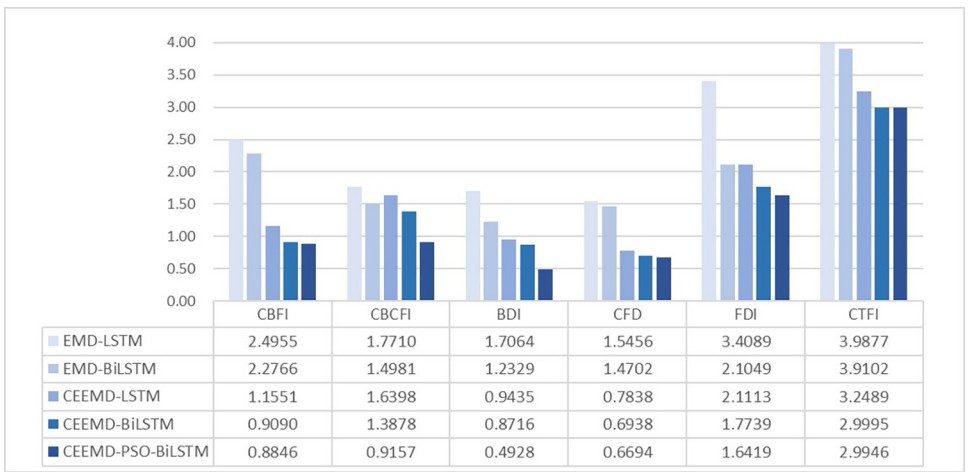

**Fig 8. Comparison of SMAPE results based on different combination forecasting methods.**

model proposed in this article is a very effective method for analyzing and forecasting highly volatile and non-linear shipping index data, and has obvious advantages over other methods.

## 6. Conclusions

Due to the non-stationary and unstructured characteristics of shipping indices, this article proposes a CEEMD-PSO-BiLSTM model through the idea of "decomposition-reconstruction-integration", and makes the modelling and forecasting of non-linear, multi-scale and highly complex time series feasible and efficient through a unique IMF restructuring approach. In this framework, the price data of shipping indices are first processed by means of Complementary Ensemble Empirical Mode Decomposition, and the prediction efficiency of the subsequent model is improved by decomposing the time series into multiple time series; the IMF series decomposed by CEEMD are reorganized, and a more efficient IMF reorganization strategy is proposed in this article, i.e. the high-frequency component and low-frequency component are identified by independent sample t-test The low-frequency components are combined into a new series LF, and the PSO-BiLSTM model forecasts are carried out simultaneously with the high-frequency components and trend terms, and the individual forecasts are summed and combined into the final forecast set; the forecast results of each series are combined to obtain the final results. In this article, six mainstream shipping indices in the Chinese shipping market (FDI, CBFI, CBCFI, BDI, CFD, CTFI) are selected to test the prediction accuracy of each model with four evaluation indicators (RMSE, SMAPE, MAE, R2), and comparing the prediction results of other eight models we find that:

(1) It can be seen from the four evaluation indicators that the IMF restructuring strategy proposed in this article significantly outperforms other IMF restructuring strategies and provides a more efficient method for other related financial time series forecasting; (2) The CEEMD-LSTM model optimized by PSO has a better prediction effect (3) The forecasting results in all six shipping indices show that the CEEMD-PSO-BiLSTM model constructed in this article outperforms other models under all evaluation indicators.

In summary, this article further improves the deep learning framework for time series forecasting, enhances the forecasting accuracy of deep learning models in time series, demonstrates that the deep learning framework still has excellent forecasting results in shipping indices other than common forecasting objects (stock market, bond market, exchange rate, etc.), and hopes to continue to advance the integration of deep learning and the field of

financial forecasting, providing future research with practical and reliable modelling solutions. However, there are still some limitations in this study, for example, this article is only based on the analysis of shipping index price data, and does not introduce factors such as the influence of macro policies of various countries and the influence of investor sentiment, which is also the focus of future improvement of the forecasting model.

## Supporting information

**S1 Data.**
(XLSX)

## Author Contributions

**Conceptualization:** Chengang Li.

**Data curation:** Ying Yan.

**Formal analysis:** Ying Yan, Han Jin.

**Funding acquisition:** Chengang Li, Xuan Wang, Yongxiang Hu.

**Methodology:** Han Jin.

**Resources:** Yongxiang Hu, Guofei Shang.

**Supervision:** Chengang Li, Yongxiang Hu, Guofei Shang.

**Writing – original draft:** Xuan Wang.

**Writing – review & editing:** Xuan Wang.

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
