## [Decision Letter · Decision Letter 0]

21 Sep 2022

PONE-D-22-19137Forecasting shipping index using CEEMD and BiLSTM modelPLOS ONE

Dear Dr. Hu,

Thank you for submitting your manuscript to PLOS ONE. After careful consideration, we feel that it has merit but does not fully meet PLOS ONE’s publication criteria as it currently stands. Therefore, we invite you to submit a revised version of the manuscript that addresses the points raised during the review process.

ACADEMIC EDITOR: Please consider all the comments in the revision and a proof reading is recommended for the re-submission.

We look forward to receiving your revised manuscript.

Kind regards,

Qichun Zhang, PhD

Academic Editor

PLOS ONE

Journal Requirements:

"NO"

"NO"

Reviewers' comments:

Reviewer's Responses to Questions

**Comments to the Author**

1. Is the manuscript technically sound, and do the data support the conclusions?

Reviewer #1: Yes

Reviewer #2: Partly

2. Has the statistical analysis been performed appropriately and rigorously? 

Reviewer #1: Yes

Reviewer #2: I Don't Know

3. Have the authors made all data underlying the findings in their manuscript fully available?

Reviewer #1: Yes

Reviewer #2: No

4. Is the manuscript presented in an intelligible fashion and written in standard English?

Reviewer #1: Yes

Reviewer #2: Yes

5. Review Comments to the Author

Reviewer #1: The article presenting a Forecasting shipping index using CEEMD and BiLSTM model. I recommend to proceed this article to publication as it meet the requirements:

1. The manuscript presented in an intelligible fashion and written in standard English.

2. The authors made all data underlying the findings in their manuscript fully available.

3. The manuscript technically sound, and do the data support the conclusions.

Reviewer #2: Basically, indexing and planning are key problem in producing and manufacturing. The paper investigated a significant topic which potential impacts. However, the method has not been proposed clearly. In particular, the motivation is weak, why the author use the proposed model structure rather than other existing models. The benefits of the proposed algorithm has not been fully reflected. The result shows the performance of the method however the comparative study is not sufficient. More comparison to the intelligent optimisation methods such as GA, PSO, should be discussed in the revised version.

6. PLOS authors have the option to publish the peer review history of their article (what does this mean?). If published, this will include your full peer review and any attached files.

Reviewer #1: No

Reviewer #2: No

---

## [Author Response · Author response to Decision Letter 0]

16 Dec 2022

Dear Editor:

Thank you for the time taken to go through our manuscript! With the help of the reviewer’ comments and suggestions, substantial revisions have been made to the manuscript. Indeed, these comments and suggestions have helped to improve our paper. Please find our detailed responses below.

1. The first reviewer's comments are as follows:

The article presenting a Forecasting shipping index using CEEMD and BiLSTM model. I recommend to proceed this article to publication as it meet the requirements:1. The manuscript presented in an intelligible fashion and written in standard English.2. The authors made all data underlying the findings in their manuscript fully available.3. The manuscript technically sound, and do the data support the conclusions.

Reply: Thank you for taking time out of your busy schedule to review this article, and thank you for affirming this article.

2. The second reviewer's comments are as follows:

Basically, indexing and planning are key problem in producing and manufacturing. The paper investigated a significant topic which potential impacts. However, the method has not been proposed clearly. In particular, the motivation is weak, why the author use the proposed model structure rather than other existing models. The benefits of the proposed algorithm has not been fully reflected. The result shows the performance of the method however the comparative study is not sufficient. More comparison to the intelligent optimisation methods such as GA, PSO, should be discussed in the revised version.

Reply: Thank you for taking time out of your busy schedule to make suggestions for this article. According to your suggestion, we have done the following:

(1) Based on the Reviewer's opinion that we lack optimization algorithm, we have learned and added an improved PSO model as the optimization model of this paper. PSO has shown its powerful optimization ability in many fields, and we have used PSO in this model to produce good results. We added sections 2.4 and 3.5 to show how we optimized the CEEMD-BiLSTM model with PSO.

(2) We apologize for the lack of motivation in previous editions of the manuscript. We have added lines 70-84 in the Introduction to illustrate the purpose and innovation points of this paper. and sincerely hope that the motivation in our new edition will be stronger.

Lines 70 to 84 are as follows:

This article adopts the idea of "decomposition, reconstruction and integration" to construct a comprehensive model -- CEEMD-PSO-BiLSTM model, aiming to analyze and predict the internal characteristics and trend of shipping index, grasp the dynamic trend of shipping market, and prevent the major risks that may be brought by shipping market. At the same time, the prediction effect of deep learning in shipping index is explored to fill the gap of neural network in the field of shipping index prediction. The contribution of this article are as follows: (1) We introduce the neural network algorithm system into the shipping market prediction, and through the current better CEEMD method, the original sequence is decomposed, which creates favorable conditions for accurately fitting the nonlinear and high noise characteristics of shipping index, and significantly reduces the difficulty of neural network prediction. (2) We choose BiLSTM model with strong generalization ability in deep learning models as the framework, and combine BiLSTM model with CEEMD model in the field of signal decomposition to build a high-precision combined prediction model based on shipping index, providing a practical and reliable modeling scheme. (3) PSO was introduced to optimize the BiLSTM model, which significantly improved the prediction accuracy of shipping index.

（3）We have revised the title and conclusion of this paper. The original title is "Forecasting shipping index using CEEMD and BiLSTM model". The revised title is "Forecasting shipping index using CEEMD-PSO-BiLSTM model". In the conclusion of this paper, we add the prediction effect of CEEMD-PSO-BiLSTM model in six shipping indexes. And updated the conclusion, we think CEEMD-PSO-BiLSTM model has better prediction effect.

Thank you again for your careful review of the article, and we would appreciate any suggestions.

 Sincerely yours

---

## [Decision Letter · Decision Letter 1]

3 Jan 2023

Forecasting shipping index using CEEMD-PSO-BiLSTM model

PONE-D-22-19137R1

Dear Dr. Hu,

We’re pleased to inform you that your manuscript has been judged scientifically suitable for publication and will be formally accepted for publication once it meets all outstanding technical requirements.

Kind regards,

Qichun Zhang, PhD

Academic Editor

PLOS ONE

Additional Editor Comments:

All the concerns have been addressed in the revised manuscript. It is ready for publication as it is.

Reviewers' comments:

Reviewer's Responses to Questions

**Comments to the Author**

1. If the authors have adequately addressed your comments raised in a previous round of review and you feel that this manuscript is now acceptable for publication, you may indicate that here to bypass the “Comments to the Author” section, enter your conflict of interest statement in the “Confidential to Editor” section, and submit your "Accept" recommendation.

Reviewer #1: All comments have been addressed

Reviewer #2: All comments have been addressed

2. Is the manuscript technically sound, and do the data support the conclusions?

Reviewer #1: Yes

Reviewer #2: Yes

3. Has the statistical analysis been performed appropriately and rigorously? 

Reviewer #1: Yes

Reviewer #2: N/A

4. Have the authors made all data underlying the findings in their manuscript fully available?

Reviewer #1: Yes

Reviewer #2: Yes

5. Is the manuscript presented in an intelligible fashion and written in standard English?

Reviewer #1: Yes

Reviewer #2: Yes

6. Review Comments to the Author

Reviewer #1: All comments to authors had been considered for all reviewers and the article accepted for publication

Reviewer #2: All concerns have been addressed in the revised manuscript. The detail of the proposed algorithm has been explained with discussion. It is ready for publication now.

7. PLOS authors have the option to publish the peer review history of their article (what does this mean?). If published, this will include your full peer review and any attached files.

Reviewer #1: No

Reviewer #2: No
